# Estimation of Soil Erosion and Sediment Yield in the Lancang–Mekong River Using the Modified Revised Universal Soil Loss Equation and GIS Techniques

**Pavisorn Chuenchum** [1] **, Mengzhen Xu** [2] **and Wenzhe Tang** [1,*]

1    Institute of Hydraulic Structures Engineering and Construction Management, and State Key Laboratory of Hydroscience and Engineering, Tsinghua University, Beijing 100084, China; huw18@mails.tsinghua.edu.cn
2    River Research Institute, and State Key Laboratory of Hydroscience and Engineering, Tsinghua University, Beijing 100084, China; mzxu@mail.tsinghua.edu.cn
*    Correspondence: twz@mail.tsinghua.edu.cn; Tel.: +86-10-6279-4324

**Abstract:** The Lancang–Mekong River basin, as an important transboundary river in Southeast Asia, is challenged by rapid socio-economic development, especially the construction of hydropower dams. Furthermore, substantial factors, such as terrain, rainfall, soil properties and agricultural activity, affect and are highly susceptible to soil erosion and sediment yield. This study aimed to estimate average annual soil erosion in terms of spatial distribution and sediment deposition by using the revised universal soil loss equation (RUSLE) and GIS techniques. This study also applied remote sensing and available data sources for soil erosion analysis. Annual soil erosion in most parts of the study area range from 700 to 10,000 t/km$^2$/y with a mean value of 5350 t/km$^2$/y. Approximately 45% of the total area undergoes moderate erosion. Moreover, the assessments of sediment deposition and erosion using the modified RUSLE and the GIS techniques indicate high sediment erosion along the flow direction of the mainstream, from the upper Mekong River to the Mekong Delta. The northern part of the upper Mekong River and the central and southern parts of the lower Mekong River are the most vulnerable to the increase in soil erosion rates, indicating sediment deposition.

**Keywords:** soil erosion; sediment yield; RUSLE; Lancang–Mekong River basin

## 1. Introduction

Soil erosion affects and challenges the world's environment and natural resources [1–7], and economic and environmental dimensions with negative impacts can affect soil erosion, further resulting in low agricultural productivity, ecological collapse and high sedimentation [6–10]. Approximately 84% of the degraded lands around the world are associated with the most relevant issues about the environment with water and wind as the main agents of erosion [7,11–13]. The average soil erosion by water is estimated to exceed 2000 t/km$^2$/y with this type of erosion mainly occurring on croplands in tropical areas [14,15]. Human activities and climate change can also be triggered at a much higher rate thus simulating erosion [8,16–22]. Soil erosion by human activities is reportedly 10–15 times faster than any natural process [23]. For instance, approximately 80% of agricultural areas around the world face high to extreme erosion, and the amount of generated sediments can worsen the turbidity of rivers and increase further the concentration of pollutants [24–26]. Moreover, soil erosion and sediment yield can affect humans and the environment severely if sediment quantity exceeds the standard measurement value of aquatic organisms.

Soil erosion is the main part of the initial process of sediment delivery to rivers; in this initial process, displaced soil particles are transformed into sediments due to the influence of an agent of

erosion. The amount of sediments can decrease the potential storage capacity of reservoirs and the performance of hydraulic structures [10,27–30]. According to Reference [31], approximately 0.5% to 1% of sediment depositions affect the annual loss of storage capacity of reservoirs around the world, indicating that most dams will likely be left with only 50% of their corresponding volumes by the 2050s. Reference [32] affirms that sediments currently occupy 40% of the reservoir storages in Asia, indicating high loss of storage capacity. These circumstances affect the long-term sustainability of water sources for hydropower dams. The supposedly low sediment yield from the trapping of dams may also cause shoreline erosion, bank erosion and loss of riparian vegetation [33–36].

Lancang–Mekong River basin, as an important transboundary river in Southeast Asia, is one of the largest rivers causing high sediment loads in Asian rivers. According to Reference [37], the average annual sediment load and the specific sediment yield in the Lancang–Mekong River basin is approximately 160 Mt/y and 200 t/km$^2$/y, respectively. The upper Mekong basin contributes approximately 50% of the amount of sediments in the Lancang–Mekong River basin [37–39]. Moreover, the Lancang–Mekong River basin is beset by soil erosion and sediment problems because of rapid socio-economic development, population growth, land deterioration and deforestation in the last 50 years, and the problem is most especially caused by the development of hydropower dams in the region [38,40–42]. Many areas are easily vulnerable to soil erosion due to the influence of rainfall, runoff and human activities. In the last few years, the Lancang–Mekong River basin has eroded at an average rate of 5000 t/km$^2$/y [33] which is a moderate erosion level, and it tends to increase in intensity continuously from climate change and land-use change. Conversely, sediment yield in the river basin is decreasing from 250 t/km$^2$/y to 209 t/km$^2$/y, because the sediment quantity is trapped by hydropower dams. Historical sediment load (1960–2013) from China to the lower Mekong River indicates clearly that the amount of sediment loads heavily decreased from 84.7 Mt/y to 10.8 Mt/y and 147 Mt/y to 66 Mt/y at Chiang Saen and Pakse stations, respectively [43].

Previous research attempted to study emphatically the sediment issue in the Lancang–Mekong River basin and some parts of the basin as a means to accumulate knowledge and information for policymakers. The study of sediments in this river can be divided into two main groups. The first group of previous research focused on the changes in sediment load from the construction and operation of dams in the upper Mekong Basin. Reference [44] considered the changing sediment load in the lower Mekong basin because of the possible effects of the cascade dams in the Lancang. Reference [45] considered the effect of sediments from the Manwan Dam in both pre-dam and post-dam stages. Reference [45] estimated the sediment load of the lower Mekong River basin by classifying the rating curve of suspended sediment concentrations obtained from adjacent stations. Reference [46] investigated the nature and magnitude of changing sediment load and their trends in the Lancang–Mekong River basin using available sediment data from 1965 to 2003. Reference [34] analysed the suspended sediment flux and the sediment supply in the lower Mekong River basin using high-frequency measurements obtained from specific stations in Vietnam. Most research in the first group reveals that the construction and operation of dams in the upper Mekong River basin affect negatively the sediment load in this river due to the trapped sediments in the reservoirs. Sediment load also appears with constantly decreasing trends. Meanwhile, the second group of previous research focused on the sediment trapping efficiency of dams in the Lancang–Mekong River basin. Reference [28] analysed and predicted the sediment trapping efficiency of reservoirs in the mainstream of Lancang River. Reference [47] developed an estimation technique for the sediment trapping efficiency of existing and planned reservoirs in the Mekong River using Brune's method. Reference [48] estimated sediment yield based on geomorphic characteristics, tectonic history and available sediment data and, subsequently, considered the cumulative sediment trapping of dams. Most research in the second group indicates that the majority of sediment loads are trapped in existing dams in the upper Mekong River basin, and they will be further trapped if planned dams are operated officially in the near future. However, most of the above studies concentrated only on sediment load data and used the trapped sediment load data of dams obtained from observation stations. Conversely,

studies on soil erosion in the Lancang–Mekong River basin requiring both field surveys and other techniques are rare.

Some studies on soil erosion apply the universal soil loss equation (USLE) in combination with GIS and remote sensing techniques to analyse the spatial distributions and patterns of soil erosion in the Lancang–Mekong River basin. The method is convenient for soil erosion analysis, because it can estimate long-term soil erosion. References [49,50] estimated soil erosion in the upper Mekong River basin in Yunnan Province using USLE and analysed spatial patterns with environmental factors. Reference [51] assessed the conserved water and soil ecosystems in Yunnan Province using remote sensing techniques. Reference [52] analysed the spatial distribution of soil erosion in north-western Yunnan (Lancang River) based on the revised universal soil loss equation (RULSE) and GIS techniques. Reference [10] estimated the impact of soil erosion on the reservoirs in Yunnan Province using USLE. Reference [53] conducted a soil loss vulnerability analysis of the Mekong River basin by applying USLE. Nonetheless, the above studies identified the limitations of the USLE model, including the development of input data for new areas to satisfy the long-term data requirements, difficulties in assessing gully erosion and large-scale areas, estimation of soil loss only and insufficient computation of sediment deposition. The RUSLE model was developed accordingly to improve the estimation of potential soil erosion. The input factors in RULSE can be used by using values from the literature or adapted for empirical and statistical data in combination with GIS software. In addition, the RUSLE results are valid in terms of estimating the risks of water erosion.

Previous studies mostly investigated the changing sediment load and the sediment trapping caused by dam construction and operation. Nonetheless, the understanding of soil erosion and soil deposition is also highly important. Soil erosion, as the main part of the sediment process, can be used to plan countermeasures for the Lancang–Mekong River basin. Previous studies also emphasized that soil erosion research should focus on the simulation of sediment erosion, but they did not consider sediment deposition. Hence, this research aimed to develop methods to calculate sediment deposition and erosion based on the RUSLE model and GIS techniques and, subsequently, evaluate the impact of soil erosion on hydropower dams in the Lancang–Mekong River basin. This study only considered suspended sediment despite the limitation of the model. In addition, the factors that can influence potential and actual soil erosion in the Lancang–Mekong River basin were also determined. The simulation period of the study covered from 2000 to 2015 depending on the available data in the analysis.

## 2. Study Area

The Lancang–Mekong River basin is a transboundary river in Southeast Asia (Figure 1). Originating from China's Qinghai–Tibet Plateau, the source of the river is located in Yushu of Qinghai Province. By its name, Lancang River represents the upper Mekong River basin in China, while the downstream part is located in Yunnan Province. Along with the river portions in Myanmar, Laos PDR, Thailand, Cambodia, and Vietnam, the lower Mekong River basin has a length of 4909 km and a coverage area of 795,000 km$^2$ [38,54,55]. The average annual water discharge is approximately 475 km$^3$ [38,44,54,56]. Thus, approximately 24% of the total area comprises the upper Mekong River basin, with contribution rates of 15% to 20% of the water flow to the Lancang–Mekong River basin. Most areas comprise complex mountains and hills and deep valleys [10,38,53]. In addition, approximately 76% of the total area is developed by major tributary systems from the lower Mekong Basin, especially Lao People's Democratic Republic (Laos PDR) [38,53]. The elevation of the basin varies from 0 to 6549 m above sea level. The different elevations have varying distributed agriculture depending on climatic zones and temperature. Moreover, various elevations of the river have water development projects, such as cascade hydropower dams, in both the mainstream and the sub-basins. Furthermore, soil erosion in the Lancang–Mekong River basin results in sediment deposition. The sediments affect the dams in all statuses (i.e., operation, under construction, and planned). The dam system of the Lancang–Mekong River basin comprises 133 dams [38,57,58] including those in the mainstream and the sub-basins.

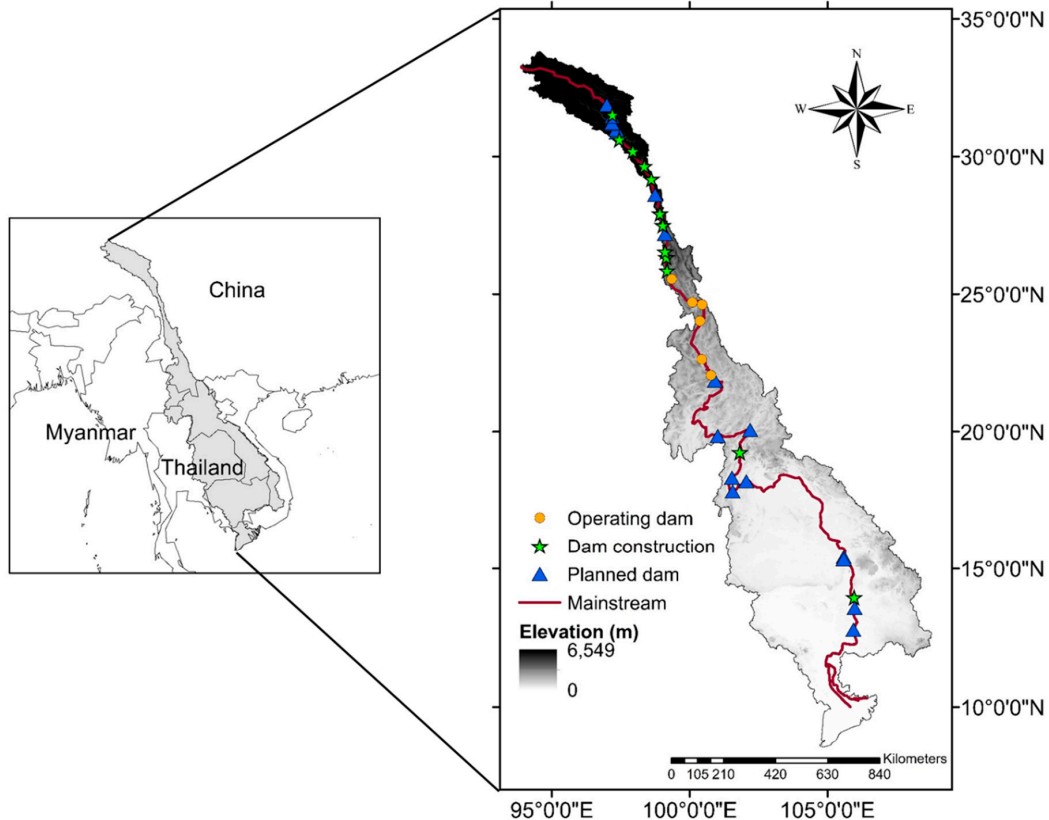

**Figure 1.** Location and elevation of the study area and location of dams in the mainstream river.

## 3. Materials and Methods

### 3.1. RUSLE

The RUSLE model, based on the USLE model, was developed by the US Department of Agriculture. The RUSLE is an empirical soil erosion model and has been recognised as a standard method to calculate the risk of average soil erosion on land. The RUSLE is also the most popular model for estimating average soil erosion in water [59], and it is simple to integrate with GIS and remote sensing [10,60–62]. Furthermore, RUSLE can provide international applicability and comparability for the results and methods, as the model can be adapted and applied in many regions globally. The RUSLE model can be expressed as follows:

$$A = R \times K \times LS \times C \times P \tag{1}$$

where:

$A$ is the mean annual soil loss (t/ha·y);

$R$ is the rainfall erosivity factor (MJ·mm/ha·hr·y);

$K$ is the soil erodibility factor (t·hr/MJ·mm);

$LS$ is the topographic factor (dimensionless);

$C$ is the cropping management factor (dimensionless); and

$P$ is the support practice factor (dimensionless).

The assessment of soil erosion in the Lancang–Mekong River basin can be classified into five levels according to the Soil Erosion Standard Document–Technological Standard of Soil and Water Conservation (SD238-87) of Reference [63].

### 3.1.1. Rainfall Erosivity Factor

Rainfall plays an important role in the process of soil erosion and sedimentation and leads to water erosion, such as splash erosion, sheet erosion, rill erosion and gully erosion, caused by water flow. Soil particles, which are transported away from a site by the flow, are those detached by rainfall impact [64]. Therefore, high-potential erosion can be determined by rainfall intensity and storm duration. Normally, the relationship between total storm energy (E) and maximum 30 min intensity (I30) can be regarded as the R factor, as reported by Reference [65]. Given the limitation of precipitation data about the river, the R factor is derived from the Asian Precipitation Highly Resolved Observational Data Integration Towards Evaluation of the Extreme Events (APHRODITE) for the period from 2000 to 2015 which also correspond to the daily gridded precipitation data for Monsoon Asia [66]. This project is developed from the daily rain gauge data for the Asia region and cover nearly 12,000 stations. This study has selected the highest fine-gridded resolution (spatial resolution of 5 km) of available precipitation data. For the conditions in the Lancang–Mekong River basin, this study chose the formula of the R factor from References [41,67] which applied the assessment of the R factor in Southern China. Equation (2) is appropriate, because the climate and area conditions in Southern China are almost uniform to those in the Lancang–Mekong River basin.

$$R = \sum_{i=1}^{12} \left( -1.15527 + 1.792 P_i \right) \tag{2}$$

where $R$ is the rainfall erosivity factor (MJ·mm/ha·hr·y), and $P_i$ is the monthly rainfall (mm).

### 3.1.2. Soil Erodibility Factor

The effect of soil characteristics and soil properties on soil erosion can be represented by the soil erodibility factor ($K$), because this factor shows the physical and chemical properties of the soil through the equations related to soil texture, soil organic matter and percentages of sand, silt, and clay. Furthermore, the $K$ factor is based on soil permeability and particle size distribution. The $K$ factor is strongly related with the $R$ factor through the soil erosion rate per kinematic energy of rainfall erosivity index. The observed data of the local soil properties in the Lancang–Mekong River basin are extremely difficult to access. Thus, the soil data in this study were derived from the SoilGrids map which is developed and maintained by ISRIC–World Soil Information. This study used the available soil data grid with a spatial resolution of 1 km. The data on soil properties were analysed using the methods in References [68,69], in which the percentages of silt, clay, sand and organic carbon fraction were calculated by Equations (3)–(6). Soil erodibility was computed according to the method in Reference [70] as shown in Equation (7). Then, the unit of the $K$ factor was transferred to the International System of Units (SI) [70]. This method is widely used for the analysis of the $K$ factor for soil properties such as soil structure and particle-size distribution [10,53,68,69,71,72].

$$f_{csand} = \left\{ 0.2 + 0.3 \exp\left[ -0.256 m_s \left( 1 - \frac{m_{silt}}{100} \right) \right] \right\} \tag{3}$$

$$f_{cl-si} = \left( \frac{m_{silt}}{m_c + m_{silt}} \right)^{0.3} \tag{4}$$

$$f_{orgC} = \left\{ 1 - \frac{0.25 orgC}{orgC + \exp[3.72 - 2.95 orgC]} \right\} \tag{5}$$

$$f_{hisand} = \left\{ 1 - \frac{0.7\left(1 - \frac{m_s}{100}\right)}{\left(1 - \frac{m_s}{100}\right) + \exp\left[-5.51 + 22.9\left(1 - \frac{m_s}{100}\right)\right]} \right\} \tag{6}$$

$$K = f_{csand} \times f_{cl-si} \times f_{orgC} \times f_{hisand} \tag{7}$$

where $K$ is the soil erodibility factor, $f_{csand}$ is the function of high-coarse sand content in soil, $f_{cl-si}$ is the function of clay and silt in soil, $f_{orgC}$ is the function of organic carbon content in soil, $f_{hisand}$ is the function of high sand content in soil, $m_s$ is the percentage of sand fraction content (particles with diameters from 0.05 to 2 mm) (%), $m_{silt}$ is the percentage of silt fraction content (particles with diameters from 0.002 to 0.05 mm) (%), $m_c$ is the percentage of clay fraction content (particles with diameters of <0.002) (%), and $orgC$ is the percentage of organic carbon content of the layer (%).

### 3.1.3. Topographic Factor

The topographic factor (*LS*) includes slope length (*L*) and slope steepness (*S*), which are the two important influencing parameters of soil erosion. Both GIS and remote sensing techniques were applied to access the *LS* factor in the RUSLE equation using the digital elevation model (DEM) [73]. For a large area, grid resolution is important for soil erosion estimation [74]. Changes in grid size affect steepness values, both directly and indirectly. The *L* factor depends on grid size and steepness, while the *S* factor affects steepness only. Hence, if the DEM data have a high resolution, then the model output can increase the accuracy of the *LS* factor in the RUSLE model [75,76]. Digital elevation model images with a 1 km resolution were downloaded from the US Geological Survey (https://earthexplorer.usgs.gov). Past researchers applied high-resolution DEM images for soil erosion determination because of these images' good accuracy and reliability [6,10,16,20,49–51,53,60,62,73,76–79]. The calculation of the *LS* factor can be based on the RUSLE principle by using the GIS software as explained in References [20,73,78,80–82]. The *S* factor was calculated in two conditions (Equations (8) and (9)), and the *L* factor was computed with Equation (10). Then, the *LS* factor in each grid cell was coupled in Equation (11).

$$S_{factor} = 10.8\sin\theta + 0.03; \text{ slope gradients} < 9\% \tag{8}$$

$$S_{factor} = 16.8\sin\theta + 0.50; \text{ slope gradients} \geq 9\% \tag{9}$$

$$L_{factor} = \left(\frac{\lambda}{22.12}\right) \times \left(\frac{\frac{\left(\frac{\sin\theta}{0.0896}\right)}{(3\sin\theta\times0.8+0.56)}}{1 + \frac{\left(\frac{\sin\theta}{0.0896}\right)}{(3\sin\theta\times0.8+0.56)}}\right) \tag{10}$$

$$LS = L_{factor} \times S_{factor} \tag{11}$$

where $\lambda$ is the length of the slope, $L_{factor}$ denotes the slope length factors, and $S_{factor}$ is the slope steepness factor.

### 3.1.4. Cropping Management Factor

Vegetation cover is one of the most important factors affecting the erosion process and the development of rivers [64]. Moreover, vegetation cover can shield the soil surface from the impact of falling rain and slow down the velocity and scouring power of runoffs. Normally, vegetation cover can be depicted by the cropping and management practices in an area through the *C* factor. The range of the *C* factor is between 1 and 0. If the *C* factor is equal to 1, then no vegetation cover (i.e., bare land) exists in that area. If the *C* factor is close to 0, then strong vegetation cover exists, indicating protection against soil erosion.

The product of remote sensing data from the Moderate Resolution Imaging Spectroradiometer (MODIS), with a cell size of 250 m in spatial resolution, was applied. The MODIS is a good choice for large-area coverages. The normalized difference vegetation index (*NDVI*) was used in this study to estimate the *C* factor following the method of [83]. The detailed equations were given by Equations (12) and (13). The MODIS' remote sensing can investigate all months, from the historical period to the present (2000–2015), to investigate the study area.

$$C = \frac{(-NDVI + 1)}{2} \tag{12}$$

$$NVDI = \frac{(NIR - RED)}{(RED + NIR)} \tag{13}$$

where *C* is the cropping management factor, *NDVI* is the average of the normalized difference vegetation index, *NIR* is surface spectral reflectance in the near-infrared band, and *RED* is the surface spectral reflectance in the red band. Both *NIR* and *RED* were extracted from the MODIS images. In reflecting the vegetation cover and the agricultural activities in the Lancang–Mekong River basin, the five months of January, April, July, October and December [38] were selected from 2000 to 2015. The average *NDVI* was calculated from these data covering 16 years.

### 3.1.5. Support Practice Factor

The support practice factor was used to express the effect of land use and land cover on soil erosion. The *P* factor describes the change in potential erosion by flowing water through the effect of supporting conservation practices such as contouring, buffer strips and terraced contour farming [6,53,65,77,84]. The maximum value of the *P* factor is usually set to 1.0 to mean no erosion control solution. A decreasing value of the *P* factor means that flowing water is reduced in terms of both volume and velocity. Moreover, a decreasing *P* also means reduced intensity of sediment deposition on the surface [85]. Given the many limitations, the *P* factor was determined on the basis of the land cover type from the *C* factor (Table 1) as suggested by [86]. Land-use type was obtained from the product of the MODIS' remote sensing with a cell size of 250 m for the spatial resolution.

**Table 1.** Land cover classification and the *C* and *P* factors [86].

| Land Cover of the RUSLE | C Factor | P Factor |
|---|---|---|
| Urban area | 0.1 | 1.0 |
| Bare land | 0.35 | 1.0 |
| Dense forest | 0.001 | 1.0 |
| Sparse forest | 0.01 | 1.0 |
| Mixed forest and cropland | 0.1 | 0.8 |
| Cropland | 0.5 | 0.5 |
| Paddy field | 0.1 | 0.5 |
| Dense grassland | 0.08 | 1.0 |
| Sparse grassland | 0.2 | 1.0 |
| Mixed grassland and cropland | 0.25 | 0.8 |
| Wetland | 0.05 | 1.0 |
| Water body | 0.01 | 1.0 |
| Permanent ice and snow | 0.001 | 1.0 |

### 3.1.6. Application of GIS Tools

The input data, such as rainfall, types of land use, and land cover, terrain and soil properties, in the RUSLE model were imported and calculated using the functions in ArcGIS 10.5. The five factors were analysed according to the spatial resolution and the coordinate system of their original data. The final results of the quantitative output of soil erosion were generated as the maximum grid with 5 km of spatial resolution depending on the original data. Soil erosion in the Lancang–Mekong River basin was analysed using the results of two types of erosion (i.e., potential soil erosion and actual soil erosion), as shown in Equation (1), in the spatial distribution. The *R*, *K*, *L*, and *S* factors were considered as potential soil erosion, whereas the *R*, *K*, *LS*, *C*, and *P* factors were examined as actual soil erosion.

### 3.2. Descriptive Statistics in the RUSLE Model

Soil erosion can be identified in each factor of the RUSLE model, indicating the influence of soil erosion on a specific area [6]. The RUSLE model is transformed into logarithmic form in Equation (15),

and multiple linear regression must be applied to examine the relationships among all factors, as shown in Equation (16), and the effects on the soil erosion rate.

$$\ln(A) = \ln(R \times K \times LS \times C \times P) \tag{14}$$

$$\ln(A) = \ln(R) + \ln(K) + \ln(LS) + \ln(C) + \ln(P) \tag{15}$$

$$\ln(A) = \beta_0 + \beta_i(\ln R) + \beta_j(\ln K) + \beta_k(\ln LS) + \beta_l(\ln C) + \beta_h(\ln P) \tag{16}$$

where $\ln(A)$ is the logarithm of soil erosion rate, $\ln(R, K, LS, C,$ and $P)$ denotes the logarithmic value of the input factors in the RUSLE model, $\beta_0$ is the intercept of soil erosion rate (constant term), and $\beta_{i-h}$ is the estimated regression coefficient of each explanatory variable. Different units of the input factors are reflected through the standard coefficient ($\beta$) in Equation (16). The factors of multiple linear regression in logarithmic form can be explained as follows: if one of the factors in the RUSLE model increases by 1% in standard deviation, then $\beta_{i-k}$ percent of the standard deviation leads to an increased value of soil erosion rate ($A$). This study sets the statistical significance level at 95% confidence in SPSS. Nonetheless, given the differences in the spatial resolutions of the input factors, some factors ($K, LS, C,$ and $P$) were estimated as 5 km ($A$ and $R$ factors) in spatial resolution using the spatially averaged values assigned in the function of ArcGIS.

### 3.3. Technique of Sediment Yield Estimation

References [20,87] proposed a new technique to estimate sediment yield or sediment deposition in each sub-basin of Thailand by modifying the original RUSLE model. They regarded the suspended sediment flow from one grid cell to the other grids as dependent on the sediment yield of the original grid cell ($S_y$) and the average sediment yield capacity of sub-basin ($S_c$). If $S_y$ is greater than $S_c$, then the sediment moves to the next site. By contrast, if $S_c$ is more than $S_y$, then the sediment is deposited. $S_y$ is calculated using the individual parameters in each grid cell (Equation (17)). In the same way, $S_c$ is calculated using the original RUSLE model with the area-averaged parameters (Equation (18)). This technique was only developed for the assessment of suspended sediment. It is not appropriate for analysing the total sediment form (i.e., bed load and suspended sediment).

$$S_y = f(I_1, I_2, \dots, I_5) \tag{17}$$

$$S_C = f\left( \frac{\sum_{i=1}^{n} I_1}{A_{ba\,sin}}, \frac{\sum_{i=1}^{n} I_2}{A_{ba\,sin}}, \dots, \frac{\sum_{i=1}^{n} I_5}{A_{ba\,sin}} \right) \tag{18}$$

$$D_i \quad \text{if } S_y < S_c \tag{19}$$

$$T_i \quad \text{if } S_y > S_c \tag{20}$$

where $S_y$ is sediment yield, $S_c$ is sediment capacity, $I_i$ represents the parameters in the RUSLE model ($R, K, LS, C,$ and $P$), $A_{basin}$ is an area of the sub-basin, $n$ is the number of data in each sub-basin, $D_i$ is the sediment deposition in a cell $i$, and $T_i$ is the sediment transportation in cell $i$. $S_y$ is the result of actual soil erosion by computing from the RUSLE input factors. $S_c$ is calculated from the summation of each parameter in the RUSLE model dividing an area of the sub-basin. The five outcomes then are multiplied as $S_c$.

The above technique can show the spatial distribution of sediment yield and sediment deposition in the Lancang–Mekong River basin, indicating an integrated consideration of the sediment issue which is the main problem for water development projects in this river. Furthermore, the technique is extremely useful in studying the influence of dam construction on sediment budget, because the loss of storage capacity of dams and the reduced transport of sediments downstream are caused by sedimentation which, in turn, is the result of soil erosion [32]. Dam design and sediment management

in operations planning can be arranged properly if the sediment budget of the river is primarily determined in dam construction.

### 3.4. Observed Sediment Data

The results from net sediment mapping or sediment deposition and erosion mapping are estimated and compared with the observed sediment data from relevant organizations, such as MRC, and the literature for verification [32,44,56,88,89]. The present study collected, from 15 stations, the average sediment load and specific sediment yield (SSY) data for each sub-basin (Figure 2) from the years 1952 to 2011 (60 years) to cover the whole basin (see Supplementary Materials) which is the time period of the data collection. Sediment loads were estimated from the suspended sediment concentration (SSC) and water discharge using the sediment rating curve, and the SSY data in the Lancang–Mekong River basin were estimated based on historical geological and geomorphological characteristics of each sub-basin [48] and historical sediment load. The results of this study will be verified with SSY in each sub-basin only. Each observational station is a representative of a sub-basin in the Lancang–Mekong River basin for verification between observed SSY (1952–2011) and estimated SSY from the modified RUSLE model (2000–2015) (see Table 4).

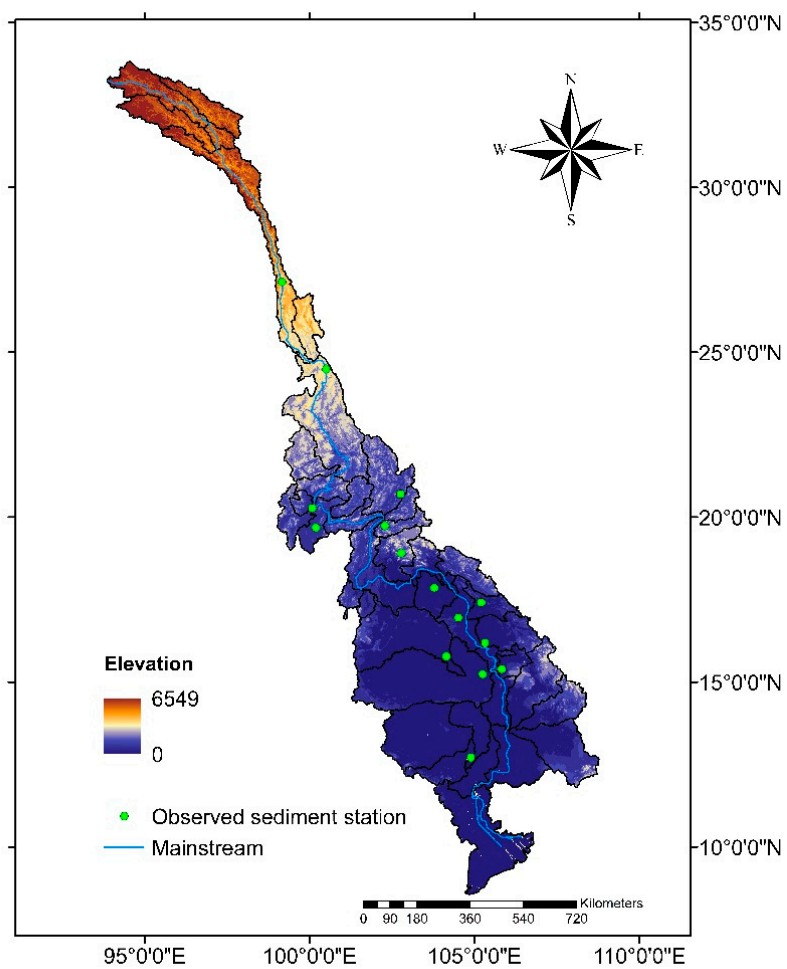

**Figure 2.** Location of sediment observational stations in the Lancang–Mekong River basin.

## 4. Results

### 4.1. Soil Erosion Factors

#### 4.1.1. Rainfall Erosivity Factor

The values of the *R* factor were analysed using Equation (2). Figure 3a shows the spatial distribution of the *R* factor for the Lancang–Mekong River basin. The range of the *R* factor was 65.6–524.3 MJ·mm/(ha·hr·y) with a mean of 294.9 MJ·mm/(ha·hr·y). The standard deviation was 80.3. The lowest values for the *R* factor were distributed mostly in the upper Mekong River basin or Lancang River in China. Meanwhile, the highest values for the *R* factor were distributed primarily in the sub-basins of Laos PDR and Cambodia and the Mekong Delta, because those areas are located along the direction of monsoon storms from the South China Sea in seasonal cycles. According to the results, the *R* factor increased from the lower basin to the upper basin, a scenario explaining the influence of climate and temperature on the river.

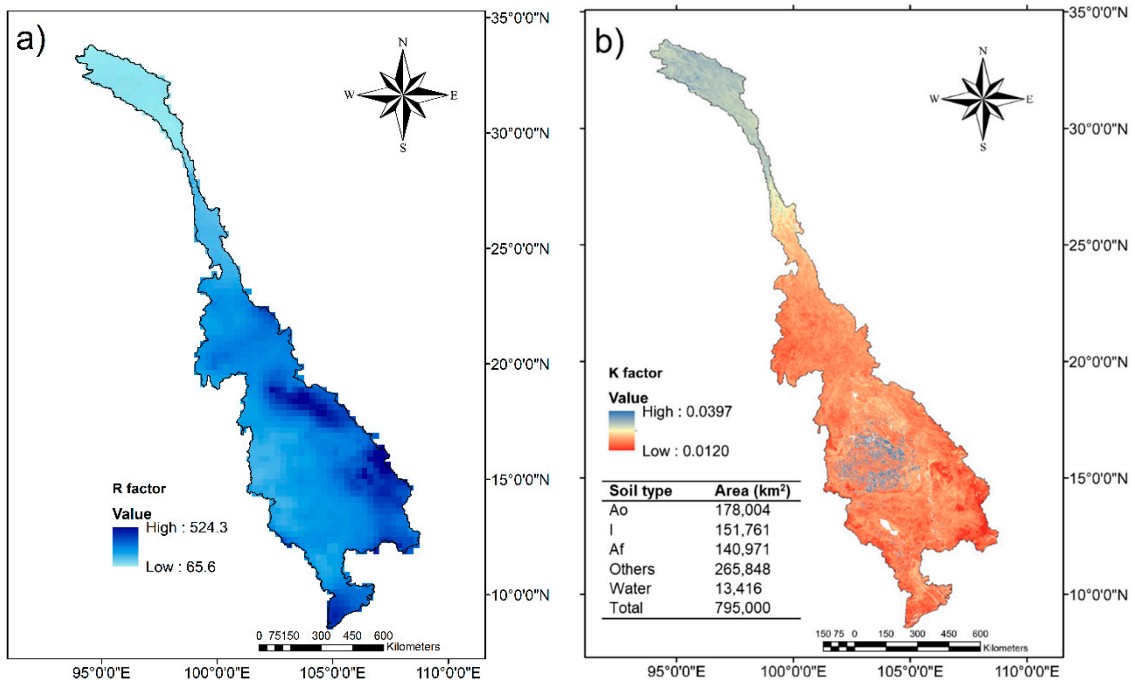

**Figure 3.** (**a**) *R* factor and (**b**) *K* factor.

#### 4.1.2. Soil Erodibility Factor

Major soil groups in the Lancang–Mekong River basin (Figure 3b) were determined using the SoilGrids database of ISRIC–World Soil Information [90,91]. The *K* factor was calculated with Equations (3)–(7). The range of the *K* factor was 0.012–0.0397 t/(hr·MJ·mm), with an average of 0.0258 t/(hr·MJ·mm). The standard deviation was 0.0012. The spatial distribution in Figure 3b indicated that the *K* factor decreased from the upper basin to the lower basin, but some areas of the Mun and Chi River basins in Thailand had high *K* values. In the Lancang–Mekong River basin, the highest elevation areas were identified by the highest *K* values, whereas the lowest elevation areas were identified by the lowest *K* values. This result corresponded with the findings in Reference [10], in which the *K* values correlated with the variation of the terrain; moreover, highly significant *K* values were found for high elevation areas such as mountains. Orthic Acrisols (Ao), Lithosoils (I) and Ferric Acrisols (Af) are the largest areas in the Lancang–Mekong River basin, and they accounted for approximately 59% of the total basin, while the other soil groups accounted for 39%.

### 4.1.3. Topographic Factor

Topographic factor was the most influential factor of soil erosion due to the flowing water from rainfall and runoff. The *LS* factor was considered from the elevation map of the Lancang–Mekong River basin (Figure 1) and the calculations of Equations (8)–(10). The range of elevation in the study area is from 0 to 6549 m above sea level, and the elevation mean was 3274 m. The basin with high elevation is mainly located in the upper Mekong River basin, and the elevation gradually decreases in the central part of the basin. More than 65% of the natural area has a slope gradient of >9%, and this area is mainly situated in the upper Mekong River basin. Slopes from 10° to 70° account for approximately 59%. Thus, the results of the *LS* factor were in the range of 0–336 (Figure 4a), and its mean value was 168. In addition, the areas represented by the *LS* values were below 60. The slope is steep, and the slope length is short. The areas with relatively high *LS* values were located in the upper part of the river, while the those with relatively low *LS* values were situated in the central part of the Mekong Delta.

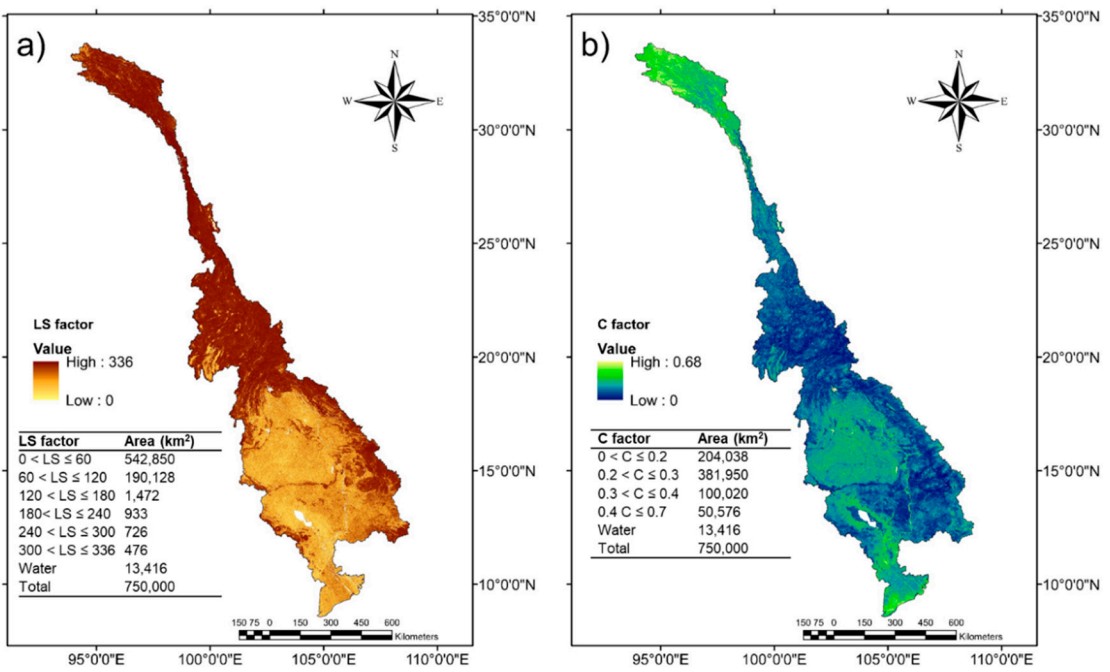

**Figure 4.** (**a**) *LS* factor and (**b**) *C* factor.

### 4.1.4. Cropping Management Factor

The *C* factor was applied using the *NDVI* analysis from the MODIS satellite images and the calculation in Equation (11). The *C* factor varied from 0 to 0.7 (Figure 4b). The *C* mean and the standard deviation were 0.34 and 0.076, respectively. Most lands in the study area are forests in parts of China, Laos PDR and Cambodia, and they were represented by relatively low values of the *C* factor. Conversely, relatively high values for the *C* factor were shown in the upper Mekong River basin in China, Thailand, and the Mekong Delta.

### 4.1.5. Support Practice Factor

The values for the *P* factor were determined following the suggestion in Reference [86] (Table 1). The change in *C* values to *P* values was applied with the functions in ArcGIS. The *P* values were 0.5, 0.8, and 1 (Figure 5). Nearly 52% of the *P* values were between 0.8 and 1, and they represent the largest portion. Thus, most areas in the basin are forests and lands with vegetation cover, indicating that soil is protected from agents of erosion. The areas with relatively high and low *P* values corresponded to similar areas for the *C* values (see Section 4.1.4).

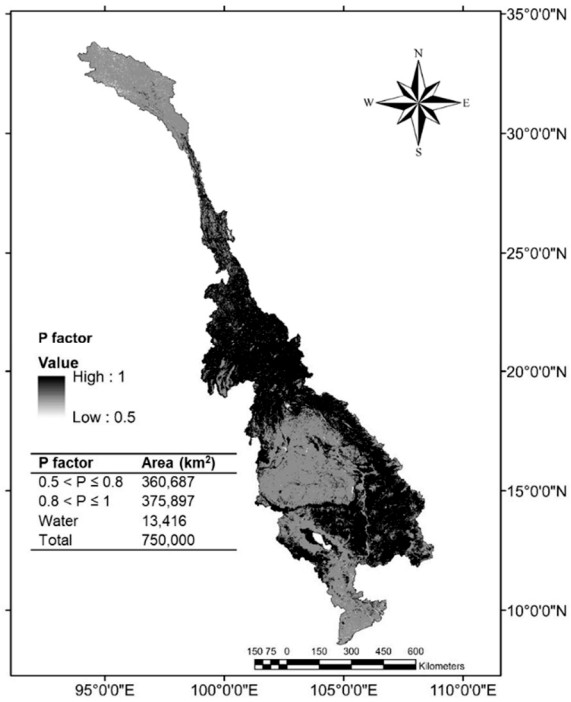

**Figure 5.** *P* factor.

## 4.2. Potential and Actual Soil Erosion

Soil erosion was divided into two types: potential and actual soil erosion. Potential erosion (*R*, *K*, *L*, and *S*) was defined as a natural erosion process without cropping management (*C*) and support practice (*P*) factors. If potential soil erosion is combined with the *C* and *P* factors, then it can be considered actual soil erosion (*R*, *K*, *LS*, *C*, and *P*). Potential soil erosion was calculated on the basis of four factors by using ArcGIS and GIS techniques. The range of potential soil erosion rate was 5000–25,000 t/km$^2$/y (Figure 6a). The average potential soil erosion was 15,000 t/km$^2$/y, and the standard deviation was 4623. The findings on spatial distribution demonstrates high-potential soil erosion in most areas in the basin. Thus, all the factors were computed for actual soil erosion (Figure 6b) which is the real-world soil erosion in the Lancang–Mekong River basin. Actual soil erosion was in the range of 700–10,000 t/km$^2$/y. The mean actual soil erosion was 5350 t/km$^2$/y, and the standard deviation was 1470. Most of the relatively high soil erosion rates were located in the north part of upper Mekong River basin and Mekong Delta. Some parts of Thailand had values close to the mean actual soil erosion. The results of the potential erosion and actual soil erosion manifested notable differences. The potential soil erosion rate was differentiated by the *C* and *P* factors because of the forest and agricultural areas. Hence, the *C* and the *P* factors play important roles in decreasing soil erosion, and they can reduce the effect by 2.5–7 times in the basin. The *C* factor indicates the capability to absorb the impact of raindrops, reduce the velocity and scouring power of runoff and reduce the runoff volume by increasing percolation into soil. Meanwhile, the *P* factor indicates the capability to decrease the amount and rate of water runoff and soil erosion with supporting cropland practices such as cross-slope cultivation, contouring farming and strip cropping.

## 4.3. Soil Erosion Risk Mapping

The results of actual soil erosion can be classified into five categories (Figure 7) according to the Soil Erosion Standard Document–Technological Standard of Soil and Water Conservation (SD238-87) [63]. Table 2 shows the soil erosion in the study area ranging from minimal erosion to extreme erosion. Most of the soil erosion in the Lancang–Mekong River basin (45% of the total area) is moderate erosion. However, the soil erosion rate is higher than 5000 t/km$^2$/y hence their classification as high erosion and

extreme erosion; the corresponding areas comprise 37% of the total area, including the high-elevation areas in China, the plateau in Thailand, Tonle Sap, and the Mekong Delta. By contrast, a low erosion rate was found mostly in Laos PDR and some parts of Cambodia because of their forest areas.

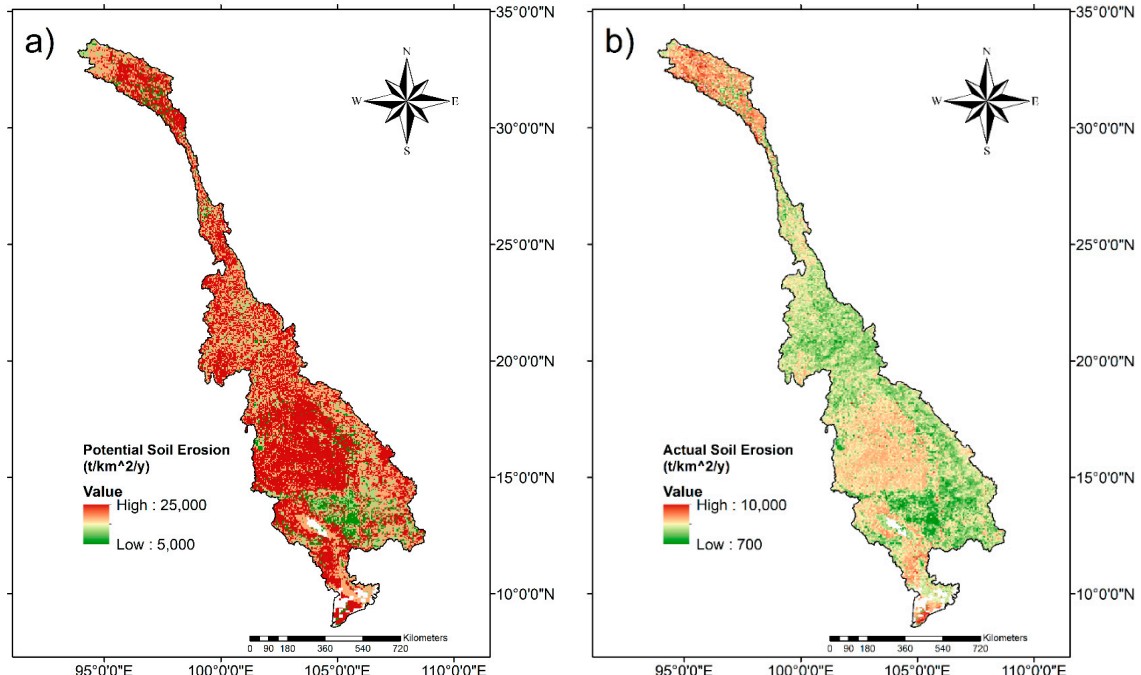

**Figure 6.** (**a**) Potential soil erosion; (**b**) Actual soil erosion.

**Table 2.** Soil erosion in the Lancang–Mekong River basin.

| Level | Soil Loss (t/km²/y) | Area (km²) | Percentage of Total Area |
|---|---|---|---|
| Minimal erosion | <500 | - | - |
| Low erosion | 500–2500 | 125,450 | 16 |
| Moderate erosion | 2500–5000 | 335,942 | 45 |
| High erosion | 5000–8000 | 253,342 | 34 |
| Extreme erosion | >8000 | 21,850 | 3 |
| Water | | 13,416 | 2 |
| Total | | 750,000 | 100 |

The analytical results on the correlation between soil erosion rate and all input factors in the RUSLE model using SPSS are shown in Table 3. The hypotheses of all factors were determined on the basis of a 95% confidence (i.e., level of statistical significance). The results were then used to build the multiple linear regression in logarithmic form for the soil erosion rate and all the RUSLE factors of the Lancang–Mekong River basin.

$$\ln(A) = 0.168 \times \ln(R) + 0.364 \times \ln(K) + 0.898 \times \ln(LS) + 0.184 \times \ln(C) + 0.246 \times \ln(P) \qquad (21)$$

Equation (21) is given by the values of standardized coefficients that are strongly related with all the RUSLE factors of the soil erosion rate. The strongest influencing factor for soil erosion in the study area was the *LS* factor ($\beta = 0.898$). Therefore, slope length and slope steepness directly affect soil erosion. In other words, soil erosion likely occurs because of gravity erosion and water erosion in an area.

**Table 3.** Standardized coefficients of factors in the RUSLE model.

| Independent Variable | Standardized Coefficient ($\beta$) | Significance |
|---|---|---|
| $\ln(R)$ | 0.168 | 0.000 |
| $\ln(K)$ | 0.364 | 0.000 |
| $\ln(LS)$ | 0.898 | 0.000 |
| $\ln(C)$ | 0.184 | 0.000 |
| $\ln(P)$ | 0.246 | 0.000 |

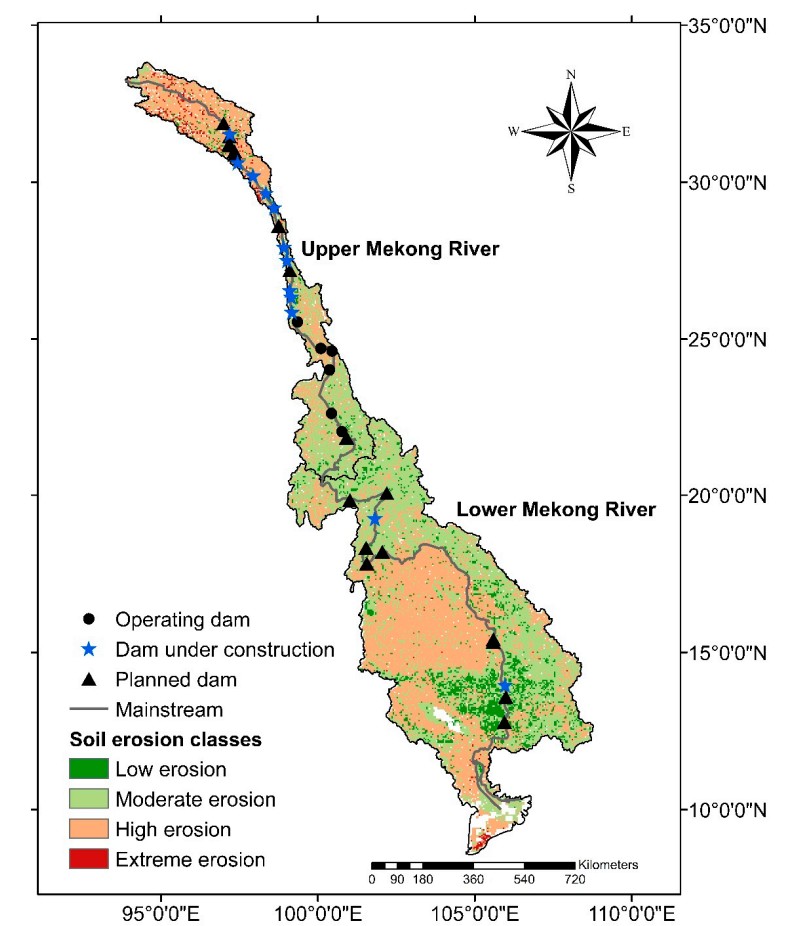

**Figure 7.** Soil erosion classification and location of dams in the Lancang–Mekong River basin.

### 4.4. Estimation of Sediment Deposition Areas

The assessment of sediment yield or sediment deposition areas in the Lancang–Mekong River basin was computed by modifying the RUSLE model according to Equations (17) and (18). The RUSLE model was determined using the spatially average parameters for the estimation of sediment yield capacity in each sub-basin. The results of the sediment yield capacity for the study area are presented in Figure 8a. Most of the sub-basins have high sediment yield capacities. Some sub-basins have low sediment yield capacity in the central part and the north part of upper Mekong River basin. The size of the sub-basin and the elevations directly result in sediment yield capacity. The average sediment yield capacity ($S_c$) was compared with the estimation of sediment yield ($S_y$) to assess the sediment deposition and sediment erosion in each grid. If the result of $S_y$ was higher than $S_c$, then sediment erosion appeared in each grid cell. Conversely, if $S_y$ was lower than $S_c$, then sediment deposition appeared in each grid cell. The results of sediment deposition and sediment erosion capacities in each grid cell are shown in Figure 8b. Sediment erosion is presented as positive values, whereas sediment deposition is presented in negative values. Potential sediment deposition and erosion are in the range from less than $-3000$ to more than 6200 t/km$^2$/y. The mean was 2105 t/km$^2$/y, and the standard deviation was 2033.

Relatively high sediment erosion occurs along the flow direction of the mainstream, including the north part of upper Mekong River basin, Laos PDR, Tonle Sap and Mekong Delta. Meanwhile, most areas in Yunnan Province, Thailand and Cambodia have high sediment depositions. The sediment deposition and erosion results can be verified using the observed sediment data from the 15 stations along the Lancang–Mekong River basin. The scatter plot of the whole basin, which was based on the observed sediment data and the assessment data on sediment yield from the RUSLE model, shows good results with a correlation higher than 0.9 (Figure 9). The RUSLE model and the technique used to assess sediment deposition and erosion can be applied in the research and prediction of soil erosion and SSY.

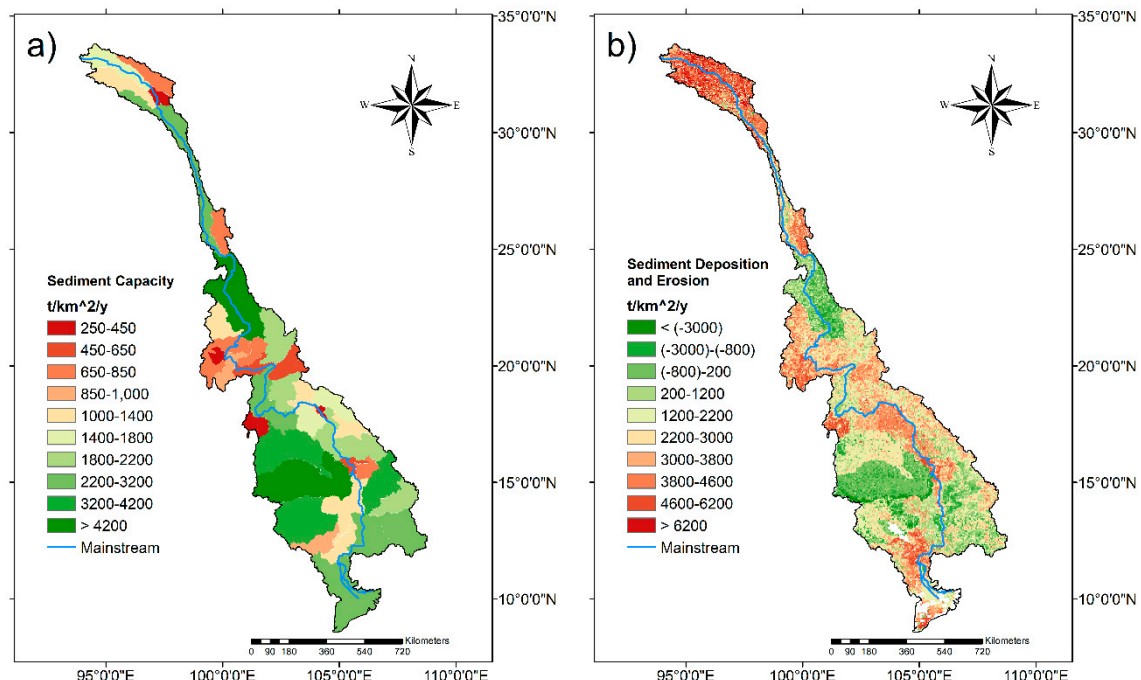

**Figure 8.** (**a**) Sediment capacity in each sub-basin, (**b**) Deposited and eroded sediments in each sub-basin.

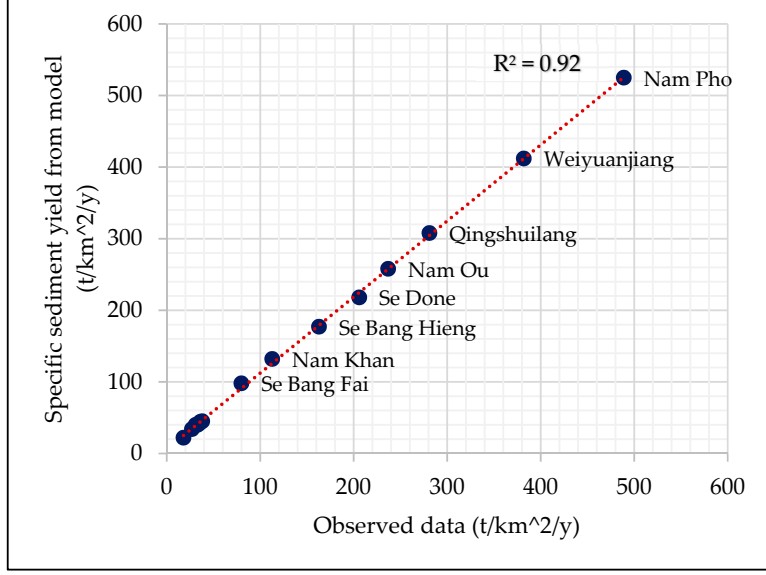

**Figure 9.** Comparative result of the whole basin based on observed data and the data of the modified RUSLE model by using the sediment estimation technique.

## 5. Discussion

### 5.1. Soil Erosion Rate in the Lancang–Mekong River Basin

This study focused on the assessment of soil erosion rate and sedimentation in the Lancang–Mekong River basin using the RUSLE model and GIS techniques with related available data. The river as the research object had many data limitations, and accessing input data to develop research on soil erosion, sediment yield capacity and sediment transport was difficult. This study attempted to utilize previous research on soil erosion in the Lancang–Mekong River basin [10,49,50,53,79], as no other evidence and information exist on how much the soil erosion rate has changed in this basin. The average soil erosion in the previous research ranges from 1400 to 8500 t/km$^2$/y. Our results are in the range near the mean values of the previous research. The spatial pattern of soil erosion occurrence in the north part of upper Mekong River basin is generally consistent with the findings of [10,49,50,53,79], but some spatial soil erosion results differ in other areas, especially in the lower part of Mekong River. Furthermore, we presumed that the different results can be attributed to the computation of the *R* factor (the main factor in soil erosion) which is influenced by differences in rainfall data. Each of the available rainfall data were previously developed for the purpose of individual projects. Nonetheless, if the *R* factor was developed from local rainfall stations in the six riparian countries, then the soil erosion rates can be regarded accurate and be further improved. Meanwhile, the results of descriptive statics in the RULSE model clearly showed that the *LS* factor is the most influential factor for soil erosion and sediment yield in the Lancang–Mekong River basin, especially in the upper Mekong River. Most studies on soil erosion and sedimentation claim that the geographical features of the Lancang–Mekong River basin, such as its steep slopes and the slope length of its hills and mountains, are affected directly by the occurrence of soil erosion in specific areas, and these sediments are transformed when transported along the river [10,32,35,41,47,49–53,79]. Consequently, the mitigation measures currently used to reduce soil erosion need to further consider solutions related with the *LS* factor such as the implementation of check dams and the application of vegetation cover. In order to consider the analytical results on the correlation between soil erosion rate and all input factors in the RUSLE model using SPSS, the *LS* factor is the strongest influencing factor on soil erosion in the study area. Nonetheless, the analytical results may not be quite effective, because the *LS* factor varies greatly in the river basin against other factors. Therefore, this section should be considered by regarding the different geological and geomorphological characteristics of the river basin such as mountains, piedmont and lowland. Moreover, different altitudinal conditions are also important conditions that directly affect the RUSLE input factors. This issue needs to be improved correctly for understanding the influencing factor on soil erosion in each feature of the river basin. In addition, the case study on potential and actual soil erosion verifies the ability of the *C* and the *P* factors to protect and reduce soil erosion. Natural vegetation covers, such as the forests in Laos PDR and Cambodia, can decrease soil erosion at rates greater than those of agricultural areas in Thailand. Hence, if forested areas are transformed into agricultural activities, then the soil erosion rate will increase remarkably, especially in upstream areas [20].

The Soil Erosion Standard Document–Technological Standard of Soil and Water Conservation (SD238-87) [63] was applied in this study to classify the soil erosion rate in the Lancang–Mekong River basin. One of the reasons is that the river has not been evaluated using the standard on soil erosion classification. Previous research used the soil erosion classification in References [63,92]. However, the number of classifications in Reference [92] is lower than that in Reference [63] and, thus, does not correspond with the results of our study. The highest value of severe erosion according to Reference [92] is greater than 3300 t/km$^2$/y, while extreme erosion according to Reference [63] is greater than 8000 t/km$^2$/y. The values differ considerably in terms of soil erosion classification. We suppose that the soil erosion classification should depend on the researcher's discretion and the suitability of research results until a set of criteria is developed by relevant credible agencies such as the Lancang–Mekong Cooperation (LMC) or the Mekong River Commission (MRC).

### 5.2. Estimation of Sediment Yield Using the Modified RUSLE Model

Soil erosion is the initial process of the sedimentation process of a river channel. The Lancang–Mekong River basin faces the challenge of sediment starvation due to the implementation of water development projects, especially hydropower dams. Most studies confirm that sediments have started to decrease continuously because of sediment trapping by hydropower dams [28,44,47,48,93]. Therefore, sediment yield capacity and sediment deposition should be analysed by relevant organizations and the six riparian country governments when drafting the needed solutions. However, the field measurement of sediment aspects is very difficult due to the limitations of equipment and nations' borders in the Lancang–Mekong River basin. Hence, the modified RUSLE model was used for the estimation of sediment yield. This method was also clearly applied to understand the sediment deposition and erosion.

The developed RUSLE model and new technique used to assess sediment yield capacity and sediment deposition areas were appropriate, and the observed sediment data and the sediment yield results from the RUSLE simulation were well correlated despite the limitation of investigating a large field survey area. The consistency between observed sediment data and the RUSLE results can also improve the accuracy of soil erosion prediction and the analyses of sediment yield capacity and sediment deposition. Nevertheless, the observed sediment data from the 15 stations were insufficient for validation, especially for the upper part of the Mekong River and all the sub-basins. This study could only access two stations (i.e., Jiuzhou and Gajiu) in the upper Mekong River basin. If other sediment data regarding the upper Mekong River can be acquired for analysis, then the effectiveness of the RUSLE model with the abovementioned technique can be effectively assessed. Besides, the results of sediment yield in some of the sub-basins may have been overestimated. Problems may have also been caused by the analysis of the RUSLE input factors which are also likely overestimated values. Additionally, the assessment of sediment deposition and erosion using the modified RUSLE model may have led to overestimated results for the sub-basins. A previous study [20] also showed the same trend for Thailand after applying the modified RUSLE model. Therefore, in the application of the method, the abovementioned limitations should be considered for model enhancement. The method proposed in this study is useful in furthering the research and analysis of sediment load at reservoirs and sediment transport in the Lancang–Mekong River basin. Furthermore, the results can be used as basis to understand the physical process of sedimentation in each sub-basin.

The result in Table 4 shows a quite good comparability of the observed and estimated SSY from the RUSLE model. Almost half of the sub-basins were in approximately 5–10% of the percentage error, while the remaining sub-basins were estimated at more than 10% from the observed values. The sub-basins have a high sediment quantity. The modified RUSLE model can be a well-known simulation. Conversely, if sub-basins have a low sediment quantity, the model shows low performance for sediment yield estimation. These causes may occur from two important factors including the spatial resolution of the RUSLE input factors and the features of the river basins. For the spatial resolution in the analysis, this study chose a rather coarse grid (5 km) resolution despite the limitations of the input data sources. The model can be well-captured in some specific areas from the influence of grid resolution. If this study can be applied to a spatial resolution of 1 km, the sediment yield estimation may be improved efficiently [20]. Meanwhile, the features of the river basins directly affect the sediment yield estimation, especially rainfall from changing climate and land-use change from human activity. Most land in the sub-basins, which have greater values of percentage error (10–29%), have changed from forest areas to agricultural areas (among other types), especially Nam Chi, Nam Mun, Nam Songkhram, and Nam Ngum. This issue created inaccuracies in the analysis of the *C* and *P* factors, because the *C* factor was considered from the MODIS satellite image using the remote sensing techniques, and the *P* factor was also estimated from the *C* factor [86]. Furthermore, sub-basins, which are overestimated values, have features without high slopes when comparing with other sub-basins. Hence, the modified RUSLE model may be able to consider areas with better slopes which is quite

consistent with Reference [20]. Totally, these factors may be the causes of the problem in the study of sediment yield estimation in the Lancang–Mekong River basin.

**Table 4.** Comparative results between observed SSY and estimated SSY from model.

| Sub-Basin | Area (km²) | Observed SSY (t/km²/y) | Estimated SSY from Model (t/km²/y) | Percentage Error (%) |
|---|---|---|---|---|
| Qingshuilang | 87,205 | 281 | 308 | 10% |
| Weiyuanjiang | 120,000 | 382 | 412 | 8% |
| Nam Pho | 184,845 | 489 | 525 | 7% |
| Nam Chi | 43,100 | 18 | 22 | 22% |
| Nam Kam | 2360 | 35 | 42 | 20% |
| Nam Khan | 5800 | 113 | 122 | 8% |
| Nam Mae Ing | 5700 | 38 | 45 | 18% |
| Nam Mun | 116,000 | 27 | 34 | 26% |
| Nam Ngum | 5220 | 36 | 44 | 22% |
| Nam Ou | 19,700 | 237 | 258 | 9% |
| Nam Songkhram | 4650 | 31 | 40 | 29% |
| Se Bang Fai | 4520 | 80 | 98 | 23% |
| Se Bang Hieng | 19,400 | 163 | 177 | 9% |
| Se Done | 5760 | 206 | 218 | 6% |
| St. Sen | 14,000 | 33 | 40 | 21% |

*5.3. Soil Erosion Impact on Dams*

Soil erosion can negatively affect hydropower dams in the Lancang–Mekong River basin. Sediments resulting from soil erosion can decrease the storage capacity of dams used to generate electricity and for other purposes. The upper Mekong basin, especially in the northern area, is classified as having extreme and high soil erosion, indicating increased vulnerability to soil erosion rate. Dams under construction and planned dams may also face the risk of increased sedimentation once they become operational. The dams located in the central and south parts of upper Mekong River basin are relatively less risky than those in the north part, because soil erosion in those areas have low and moderate erosion levels. Previous studies [10,53,79] obtained results that coincide with our research for the analysis of soil erosion impact on dams in the upper Mekong River basin. Meanwhile, soil erosion in the lower Mekong River basin, especially from the Khorat Plateau (Thailand) to the Mekong Delta, can also generate sedimentation problems due to the high occurrences of soil loss. The agricultural activities in these areas mainly cause the increase in the soil erosion rate. A dam under construction (Don Sahong) and four planned dams (Ban Koum, Phu Ngoy, Stung Treng and Sambor) may be threatened by soil erosion due to the impact of sub-basins in the Khorat Plateau in Thailand, particularly the confluence with the Lancang–Mekong River's mainstream. In addition, extreme soil erosion occurs in the Mekong Delta. Many studies affirm that the Mekong Delta is the most vulnerable area in terms of risk of soil loss [33–36].

The impact of soil erosion on dams in the Lancang–Mekong River's mainstream can be analysed in two parts based on the water sources of the river, namely, the upper Mekong River (with three river areas from Lancang basin) and the lower Mekong River (composed of the northern highlands, Khorat Plateau, Tonle Sap and Mekong Delta). The upper Mekong River basin covers 180,000 km² or approximately 24% of the study area, while the lower Mekong River basin covers 570,000 km² or approximately 76%. The soil erosion modulus of the upper Mekong River basin is 235.7 t/km²/y. Its percentage relative to total soil erosion is approximately 36%, even if this area is smaller than the lower Mekong River basin. The soil erosion modulus of the lower Mekong River basin is 198.2 t/km²/y which represents approximately 64% of the total occurrence of soil erosion. The results of the soil erosion modulus can be explained as that the reservoirs located in the upper Mekong River basin are more vulnerable from soil erosion and increased sediment. Consequently, dams are likely to be at risk

of decreasing storage capacity continually. Our results are consistent with the findings of past studies on the impact of soil erosion on dams and sediment trapping. For instance, Reference [47] reported that the sediment trapping rates of dams under construction and the planned dams in the Lancang–Mekong River basin will increase from 51% to 69% due to the high heterogeneity of specific sediment yield in the different parts of the basin, and much higher trapped sediment load is predicted because of soil erosion resulting from socio-economic development. More than 50% of the sediment load (approximately 140 Mt) in the Lancang–Mekong River basin is expected to be trapped annually. Furthermore, more than 60% of the sediment load originates from China's end of the Lancang–Mekong River's mainstream. Existing dams, dams under construction and planned dams are expected to have the highest impact on storage capacity due to the fact of sediment load. Reference [28] reported that the main dams in the Lancang River, such as Manwan, Gongguoqiao, Dachaoshan and Jinghong, have sediment trapping rates between 30% and 70% because of the high sediment yield in the Lancang–Mekong River's mainstream and sub-basins. The storage capacity of reservoirs will continuously decrease from the sediment load due to the soil erosion in the reservoir upstream.

*5.4. Delineating Sediment Form*

This study endeavoured to estimate the sediment yield by considering factors such as soil erosion, gully erosion and rill erosion. These erosions are not the only sources of sediment into the river channel, because sediment yield is fundamentally controlled by climatic conditions, geomorphologic characteristics and anthropogenic forcing [22,48]. Some sediments are formed by erosion in the river channel. Our analysis did not take other factors into account in this study. This study could not consider erosion in the river channel due to the limitation of the modified RUSLE model which solely analyses erosion on land. For the study of channel deformations and changing river morphology, a hydrodynamic model is needed. Besides, sediment data (suspended and bed load sediment) for the Lancang–Mekong River basin are insufficient, because a number of measuring stations continue to be unavailable. This is the main limitation for further study in the basin. The results in this research can be considered together with erosion in the river channel using a hydrodynamic model; it would be able to show the sediment process on both land and in the river.

## 6. Conclusions

The RUSLE model was integrated with GIS techniques in this study to assess soil erosion and sediment yield in the Lancang–Mekong River basin. The impact of soil erosion on hydropower dams was also considered. The findings indicate that soil erosion occurs in all areas of the Lancang–Mekong River basin, accounting for 5350 t/km$^2$/y of its average soil erosion rate or approximately 45% of the basin. The north part of the upper Mekong River basin and some parts of Thailand have higher terrains than the other areas, and they have good vegetation cover and support practice. Furthermore, the *LS* factor showed that this factor was the strongest influencing factor for soil erosion in the study area. The spatial distribution of soil erosion also indicated that the norther part of the upper Mekong River basin and the central and southern parts of the lower Mekong River basin are the most vulnerable areas in terms of increased soil erosion rates due to the movement of sediments to the river. Hence, the dams in this river are highly threatened by sediment problems.

The value of pursuing research on the sediment capacity of each sub-basin of the Lancang–Mekong River basin are summarized as follows. The size of the sub-basins and their elevation directly affect the sediment capacity of the river. Moreover, the spatial distribution of sediment deposition and erosion indicates that relatively high sediment erosion occurs along the flow direction of the mainstream, from the northern part of upper Mekong River basin to the Mekong Delta. The findings on sediment yield estimation from the modified RUSLE model and the observed sediment data were in good agreement and had high correlation. The proposed technique can be applied in the assessment of sediment yield capacity and sediment deposition in the Lancang–Mekong River basin.

The modified RUSLE method was successfully applied to the assessment of the amount and spatial distribution of soil erosion and sediment deposition in the Lancang–Mekong River basin. The method can be applied not only to this river but also to other important areas. This study can help policymakers and relevant organizations improve their decision making based on the provided valuable information on soil erosion and sedimentation in this region.

**Supplementary Materials:** The following are available online at http://www.mdpi.com/2073-4441/12/1/135/s1, Table S1. Lists and location of hydropower dams and reservoirs in Lancang-Mekong River's mainstream, Table S2. Average observed SL and SSY from 1962 to 2010 based on catchment area of the station, Table S3. Data sources for the analysis of the RUSLE factors in this study, Figure S1. Locations of the sediment load observed stations.

**Author Contributions:** Conceptualization, P.C., M.X. and W.T.; Methodology, P.C., M.X. and W.T. Software, M.X. and W.T.; Formal analysis, P.C.; Investigation, M.X. and W.T.; Resources, P.C. and W.T.; Data curation, P.C. and W.T.; Writing—original draft preparation, P.C.; Writing—review and editing, M.X. and W.T.; Visualization, P.C.; Supervision, M.X. and W.T.; Funding acquisition, M.X. and W.T. All authors have read and agreed to the published version of the manuscript.

**Funding:** This research was funded by the National Natural Science Foundation of China (Grant Nos. 51579135, 51379104 and 51079070), the State Key Laboratory of Hydroscience and Engineering (Grant Nos. 2013-KY-5 and 2015-KY-5), the Chinese Academy of Sciences (XDA23090401) and the National Key Research and Development Program of China (2016YFC0402407).

**Acknowledgments:** Our sincerest appreciation to the Lancang–Mekong Cooperation and the Mekong River Commission for providing us the observed sediment data. We would also like to thank the APHRODITE Project for allowing us the use of the precipitation product. We are also thankful for the technical recommendations of Prem Rangsiwanichpong and the encouragement from my beloved wife, Usa Chuenchum.

**Conflicts of Interest:** The authors declare no conflict of interest.

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
