# Peer review of "Estimation of Soil Erosion and Sediment Yield in the Lancang–Mekong River Using the Modified Revised Universal Soil Loss Equation and GIS Techniques"

_water, doi:10.3390/w12010135_

Round 1

Reviewer 1 Report

The paper is too long but the main point is that there are no new scientific finding.  It explains a well assested prosedure. Much more scientific work is needed in order to be published.

Author Response

Thank you very much for your kind reviewer’s comments. We follow your recommendations regarding our paper, and this paper has added some contents for improving scientific work, which can be summarized as follows:

Firstly, we add the discussion regarding the correlation between soil erosion rate and all input factors in the RUSLE model by using the SPSS program. The results show that the LS factor is the strongest influence factor of soil erosion in the study area. Nonetheless, the analytical results may not be quite effective because the LS factor varies greatly in the river basin against other factors. Therefore, this section should be considered by regarding the different geological and geomorphological characteristics of the river basin such as mountains, piedmont, and lowland. Moreover, different altitudinal conditions also are important conditions that affect directly the RUSLE input factors. This issue needs to be improved correctly for understanding the influence factor of soil erosion in each feature of the river basin.

Second, the discussion of the results from the comparison between observed data and model is expanded for more explanations from the issue of percentage error between observed data and model. The result in Table 4 shows quite a good comparability of the observed and estimated SSY from the RUSLE model. Almost half of the sub-basins are approximately 5-10% of percentage error while the remaining sub-basins are estimated at more than 10% from observed values. Sub-basins have a high sediment quantity. The modified RUSLE model can be a well-known simulation. Conversely, if sub-basins have a low sediment quantity, the model shows low performance for sediment yield estimation. These causes may occur from two important factors, including the spatial resolution of the RUSLE input factors and the features of the river basins. For the spatial resolution in analysis, this study chooses rather the coarse grid (5 km) resolution despite the limitations of input data sources. The model can be well-captured in some specific areas from the influence of grid resolution. If this study can be applied to the spatial resolution of 1 km, the sediment yield estimation may be improved efficiently [20]. Meanwhile, the features of the river basins affect directly the sediment yield estimation, especially rainfall from changing climate and land-use change from human activity. Most lands of sub-basins, which are the greater values of percentage error (10-29%), have been changed from the forest areas to the agricultural areas and others, especially Nam Chi, Nam Mun, Nam Songkhram, and Nam Ngum. This issue causes the analysis of the C and P factors to be inaccurate because the C factor was considered from the MODIS satellite image by using the remote sensing techniques, and the P factor was also estimated from the C factor [85]. Furthermore, sub-basins, which are overestimated values, have the features without high slopes when comparing with other sub-basins. Hence, the modified RUSLE model may be able to consider areas with better slopes, which is quite consistent with [20]. Totally, these factors may be the causes of the problem in the study of sediment yield estimation in Lancang-Mekong River basin.

Lastly, this paper is added regarding the understanding of sediment sources because this paper mainly focuses on the sediment from soil erosion, gully erosion, and rill erosion. These erosions are not only sources of sediment into the river channel because sediment yield is fundamentally controlled by climatic conditions, geomorphologic characteristics, and anthropogenic forcing [22, 47]. Some sediments are formed by erosion in the river channel. Our analysis does not take other factors into account in this study. However, this study cannot consider erosion in the river channel despite the limitation of the modified RUSLE model that analyses solely erosion on the land. For the study of channel deformations and changing rivermorphology, the hydrodynamic model is needed. Besides, the sediment data in (suspended and bed load sediment) Lancang-Mekong River basin is insufficient because the number of measuring stations is not available. This is the main limitation for further study in the basin. The results in this research can be considered together with the erosion in the river channel by using the hydrodynamic model. It is able to show the sediment process of both land and river.

Furthermore, this paper has the theoretical contributions, which can be explained as follows:

Firstly, the problem of soil erosion and sediment in Lancang-Mekong River basin is a hot and interesting issue in this region. The assessment of soil erosion in the field is very difficult because this river basin is a large area (795,000 km2), including six riparian countries. Hence, the application of the RUSLE model is very useful for considering soil erosion. Moreover, sediment data in this basin is complicated in order to follow and monitor in planning and management. This study attempt to find new techniques for estimating sediment yield for helping and supporting the policymakers and interested persons to mitigate some problems in the near future.

Next, the consideration of soil erosion in Lancang-Mekong River basin needs to apply the local data and information from six riparian countries, but accessing data has many limitations. This study applied reliable data sources for assessment of soil erosion such as the rainfall from APHRODITE, soil properties from ISRIC, DEM from USGS, and NDVI from MODIS satellite image. These data sources are freely downloaded data. Our results showed clearly that these data sources can be a good performance in the assessment of soil erosion in Lancang-Mekong River basin. They can be applied to help increasingly the policymakers for monitoring and supporting the local data together.

Third, the RUSLE model can be modified in order to estimate sediment yield. This technique is very useful because the assessment of sediment yield is very difficult in implementation. Therefore, it needs to find some methods to help and improve the information. The modified RUSLE model uses only five input factors of the model, which is easily analyzed and save the budget more.

Fourth, this research does not only study soil erosion and sediment yield but also study the sediment deposition and erosion by using the modified RUSLE model with GIS techniques. We can know that which are sub-basins to have high sediment quantity or low sediment quantity. This research can further study in sediment starvation if it is also considered with sediment trapping from dams. Most research study only soil erosion in some parts in Lancang-Mekong River basin.

We believe that this research is a valuable study in soil erosion and sediment for helping the policymakers and relevant others to plan and manage the decision making efficiently about this basin.

Reviewer 2 Report

The topic you address is very interesting, as well as challenging. In my opinion, although the methods you use have been used repeatedly in literature, the contribution of your study in the topics of hydromorphology and soil erosion is beyond any doubt. The level of English is high, as is the scientific level of your text.

However, this Reviewer has a few comments and questions.

Line 244: The period from 2000 to 2015 is a 160-year period and not a 15-year period, as stated.

From what I have understood, the simulation period (for soil erosion) is from 2000 to 2015. However, I did not find this clearly stated anywhere in the text. Please clearly state (wherever in the text you think more fitting) the period for which the results are presented.

Lines 263 – 264: While it is clear (to me) what you mean, I kindly suggest that you rewrite the sentence, highlighting that what differentiates actual erosion from potential erosion is the factors C and P of RUSLE, which introduce the counter erosion effect.

Lines 285 – 288: The exact same reasoning, for the calculation of net erosion (net erosion = gross erosion – deposition) is used in paragraph 3.1.2 of the following study:

Kaffas, K., Hrissanthou, V., Sevastas, S., 2018. Modeling hydromorphological processes in a mountainous basin using a composite mathematical model and ArcSWAT. Catena, 162: 108-129.

You may want to refer to this in order to increase the robustness of your choice. This is only a kind suggestion and it is totally up to your discretion.

Lines 291, 292: In Equations (17) and (18), both Sy and Sc are functions of I1, I2,…, I5.

To my understanding, Sy and A (from Equation (1)) should be the same thing. Is this correct? If so, please mention it, so that it is clear to the reader. Also, in this case, it is clear how Sy is calculated.

However, as in the original study (Rangsiwanichpong et al., 2018), it is not explained how Sc is calculated. The only information provided is that Sc is a function of I1, I2,…, I5 and the sub–basin area (A). If it is possible, could you provide some additional information on how Sc is calculated?

Lines 293, 294: Please use the same spacing in Equations (19) and (20)

Line 310: Please replace “observed station” with “observational station” or “measuring station”. Please to the same in Figure 2, below.

Lines 307 and 446: In these lines, you mention that your results were compared with observed data from 15 measuring stations throughout the basin. In fact, this is one of the strong elements of your study, as this kind of data is very rare in literature. Most often, if observed sediment data is available, this is in the form of sediment discharge or sediment yield at some stream segment or at the basin outlet. Hence, I suggest that you highlight this.

Due to the importance and rarity of such data, I strongly suggest that you add an additional sub-section (possibly in “Materials and Methods”), entitled “Observed Sediment Data”, in which you provide detailed information (measuring methods used, the period of record, duration of measurements, etc.) about this data. If this is not possible, please provide some general information throughout the text.

Figures 6 and 8: These figures are of poor resolution, please improve. In terms of resolution, there is room for improvement in all figures in the manuscript (apart from Figure 9).

Table 4: In what period does the estimated SSY refer to? Are these values average annual values obtained from the period 2000-2015? More importantly, what is the period which observed SSY refer to? Please clearly state this.

This research is a quality work that could interest a broad readership from such disciplines as soil erosion, hydrology, geography and remote sensing. Moreover, it fits into the scope of WATER. This Reviewer recommends the publication of this manuscript after moderate revision.

Good luck

Author Response

Thank you very much for your kind review of our manuscript. We revised it according to your kind recommendations.

Point 1: Line 244: The period from 2000 to 2015 is a 16-year period and not a 15-year period, as stated. From what I have understood, the simulation period (for soil erosion) is from 2000 to 2015. However, I did not find this clearly stated anywhere in the text. Please clearly state (wherever in the text you think more fitting) the period for which the results are presented.

Response 1: We revised a sentence from 15 years period to 16 years period in line 253. Besides, we add the simulation period of this study in line 123-124 as follows:

The simulation period of the study covers from 2000 to 2015 depending on the available data in analysis.

Point 2: Lines 263 – 264: While it is clear (to me) what you mean, I kindly suggest that you rewrite the sentence, highlighting that what differentiates actual erosion from potential erosion is the factors C and P of RUSLE, which introduce the counter erosion effect.

Response 2: We rewrite this sentence as follows:

The R, K, L, and S factors are considered as potential soil erosion, whereas the R, K, LS, C, and P factors are examined as actual soil erosion.

Point 3: Lines 285 – 288: The exact same reasoning, for the calculation of net erosion (net erosion = gross erosion – deposition) is used in paragraph 3.1.2 of the following study:

Kaffas, K., Hrissanthou, V., Sevastas, S., 2018. Modeling hydromorphological processes in a mountainous basin using a composite mathematical model and ArcSWAT. Catena, 162: 108-129.

You may want to refer to this in order to increase the robustness of your choice. This is only a kind suggestion and it is totally up to your discretion.

Response 3: We add this reference according your recommendations to increase the robustness of our manuscript in line 293.

Point 4: Lines 291, 292: In Equations (17) and (18), both Sy and Sc are functions of I1, I2,…, I5.

To my understanding, Sy and A (from Equation (1)) should be the same thing. Is this correct? If so, please mention it, so that it is clear to the reader. Also, in this case, it is clear how Sy is calculated.

However, as in the original study (Rangsiwanichpong et al., 2018), it is not explained how Sc is calculated. The only information provided is that Sc is a function of I1, I2,…, I5 and the sub–basin area (A). If it is possible, could you provide some additional information on how Sc is calculated?

Response 4: We add the explanations regarding the calculations of Sy and Sc in line 306-311 according to your kind recommendations as follows:

where Sy is sediment yield, Sc is sediment capacity, Ii represents the parameters in the RUSLE model (R, K, LS, C, and P), Abasin is an area of the sub-basin, n is the number of data in each sub-basin, Di is the sediment deposition in a cell i, and Ti is the sediment transportation in cell i. Sy is the result of actual soil erosion by computing from the RUSLE input factors. Sc is calculated from the summation of each parameter in the RUSLE model dividing an area of the sub-basin. The five outcomes then are multiplied as Sc.

Besides, A in Equation (1) is the mean annual soil loss but A in Equation (18) is the sub-basin. Thus, we would like to change A in Equation (18) to Abasin for preventing confusion.

Point 5: Lines 293, 294: Please use the same spacing in Equations (19) and (20).

Response 5: We revised the Equation (19) and (20) completely.

Point 6: Line 310: Please replace “observed station” with “observational station” or “measuring station”. Please to the same in Figure 2, below.

Response 6: We revised line 310 and the description under Figure 2 completely to be observational station.

Point 7: Lines 307 and 446: In these lines, you mention that your results were compared with observed data from 15 measuring stations throughout the basin. In fact, this is one of the strong elements of your study, as this kind of data is very rare in literature. Most often, if observed sediment data is available, this is in the form of sediment discharge or sediment yield at some stream segment or at the basin outlet. Hence, I suggest that you highlight this.

Due to the importance and rarity of such data, I strongly suggest that you add an additional sub-section (possibly in “Materials and Methods”), entitled “Observed Sediment Data”, in which you provide detailed information (measuring methods used, the period of record, duration of measurements, etc.) about this data. If this is not possible, please provide some general information throughout the text.

Response 7: observed sediment data is not available for providing full version, but we provide average sediment load and specific sediment yield of 15 stations in each sub-basin, which is in the supplementary materials.

Furthermore, we add sub-section “Observed Sediment Data” in line 320-332 according to your kind recommendations. For detail information in this study, we cannot know the measuring methods. However, we know that this dataset was collect from 1952 to 2011, and sediment load was calculated from the suspended sediment concentration (SSC) and the specific sediment yield (SSY) was considered from historical geological and geomorphological characteristics of each sub-basin and historical sediment load.

Point 8: Figures 6 and 8: These figures are of poor resolution, please improve. In terms of resolution, there is room for improvement in all figures in the manuscript (apart from Figure 9).

Response 8: We revised these figures completely.

Point 9: Table 4: In what period does the estimated SSY refer to? Are these values average annual values obtained from the period 2000-2015? More importantly, what is the period which observed SSY refer to? Please clearly state this.

Response 9: We add the explanations in line 330-332 as follows:

Each observational station is a representative of a sub-basin in Lancang-Mekong River basin for verification between observed SSY (1952-2011) and estimated SSY from the modified RUSLE model (2000-2015).

Reviewer 3 Report

The presented study is interesting, and its results are relevant for one of the most populated regions of the world. The manuscript is logically constructed and well designed. The research methodology is fully disclosed for understanding at this scale of the study. Along with this, I have a number of small comments and suggestions for the authors.

1. Soil, rill and gully erosion are not the only sources of sediment in rivers. Some sediments are formed, especially in favorable geological and geomorphological conditions, due to erosion in river beds due to vertical and horizontal channel deformations. I understand that the authors did not take this into account in their study. But this issue must necessarily be somehow covered in the discussion. It would be desirable to raise this issue using examples from the Mekong River basin or environmentally similar conditions in neighboring regions of Asia.

2. What type of river sediment did you analyze in your work: suspended, bedload, or totally suspended + bedload sediments? Please explain this in the manuscript.

3. Table 3, and lines 423–426. The authors indicate that “The strongest influence factor of soil erosion in the study area is the LS factor (β = 0.898). Therefore, slope length and slope steepness directly affect soil erosion.” It seems to me that such a general estimate for the entire river basin is not quite effective, since the LS factor really strongly affects the rate of erosion, and it varies greatly in the river basin against the background of other factors. It would be more correct to make this analysis (the influence of various factors) for different geomorphological conditions (mountain sector, piedmont (including high uplands) and lowland sector of the river basin). Under these different altitudinal conditions, the influence of various factors may give a different ratio than what is indicated for the entire river basin in Table 3. Thus, for example, the lowlands of the region are composed of sufficiently deep weathering crusts of fine particle size distribution, which, all other things being equal, are very favorable both for soil, rill and gully erosion, and erosion during horizontal channel deformations.

4. Please almost everywhere in the text of your manuscript use the “Lancang – Mekong River basin” instead of “Lancang – Mekong River”, otherwise the phrases “Most of the soil erosion in Lancang – Mekong River ...”, “the local soil properties in Lancang – Mekong River ”,“ In Lancang – Mekong River, the highest elevation areas are identified by the highest K values ... ”,“ ... the agricultural activities in Lancang – Mekong River ...”, etc. are extremely incorrect.

5. In the Conclusion, it is advisable to give SPECIFIC practical recommendations in a more concise form, which follow from the results of your study.

6. Table 4. For almost half of the analyzed sub-basins, you obtained a quite good comparability of the observed and calculated SSY-values (10% or less). For others, these values are greater, up to 29%. If it is possible, try to explain the reasons for the relatively large discrepancies in the second case. Maybe this is somehow associated with the environmental features of these river basins (Nam Songkhram, Nam Mun, Se Bang Fai, Nam Ngum, etc.)?

7. Table 2. For the level “Water”, soil loss is more than 8000? I think you had this in mind for the level of “Extreme erosion”.

8. It would be interesting for the reader to learn (a few sentences in the Introduction) how (quantitative information) the rates of erosion and sediment yield in the Mekong River basin have changed over the past decades due to changes in climate and land use. This also directly determines the relevance of your research.

9. There are several typos in the text. For example, line 547 “... that Mekong Delta is the most vulnerable rea in terms of risk of soil loss …”. Please check the typos everywhere in the text.

Author Response

Thank you very much for your kind reviewer’s comments. We revised our paper according to your recommendations.

Point 1: Soil, rill and gully erosion are not the only sources of sediment in rivers. Some sediments are formed, especially in favourable geological and geomorphological conditions, due to erosion in river beds due to vertical and horizontal channel deformations. I understand that the authors did not take this into account in their study. But this issue must necessarily be somehow covered in the discussion. It would be desirable to raise this issue using examples from the Mekong River basin or environmentally similar conditions in neighbouring regions of Asia.

Response 1: We add this issue in discussion 5.4 in line 610-623 as follows:

5.4 Delineating Sediment Form

This study endeavours the estimation of sediment yield by considering factors mainly from soil erosion, gully erosion, and rill erosion. These erosions are not only sources of sediment into the river channel because sediment yield is fundamentally controlled by climatic conditions, geomorphologic characteristics, and anthropogenic forcing [22, 47]. Some sediments are formed by erosion in the river channel. Our analysis does not take other factors into account in this study. However, this study cannot consider erosion in the river channel despite the limitation of the modified RUSLE model that analyses solely erosion on the land. For the study of channel deformations and changing rivermorphology, the hydrodynamic model is needed. Besides, the sediment data in (suspended and bed load sediment) Lancang-Mekong River basin is insufficient because the number of measuring stations is not available. This is the main limitation for further study in the basin. The results in this research can be considered together with the erosion in the river channel by using the hydrodynamic model. It is able to show the sediment process of both land and river.

Point 2: What type of river sediment did you analyse in your work: suspended, bedload, or totally suspended + bedload sediments? Please explain this in the manuscript.

Response 2: In this manuscript, river sediment is considered only the suspended sediment. We add some sentences in this manuscript in line 120-121 and line 298-300.

Line 120-121: This study is only considered the suspended sediment despite the limitation of the model.

Line 298-300: This technique was only developed for the assessment of suspended sediment. It cannot be appropriate to analyse the total sediment form (bed load and suspended sediment).

Point 3: Table 3, and lines 423–426. The authors indicate that “The strongest influence factor of soil erosion in the study area is the LS factor (β = 0.898). Therefore, slope length and slope steepness directly affect soil erosion.” It seems to me that such a general estimate for the entire river basin is not quite effective, since the LS factor really strongly affects the rate of erosion, and it varies greatly in the river basin against the background of other factors. It would be more correct to make this analysis (the influence of various factors) for different geomorphological conditions (mountain sector, piedmont (including high uplands) and lowland sector of the river basin). Under these different altitudinal conditions, the influence of various factors may give a different ratio than what is indicated for the entire river basin in Table 3. Thus, for example, the lowlands of the region are composed of sufficiently deep weathering crusts of fine particle size distribution, which, all other things being equal, are very favourable both for soil, rill and gully erosion, and erosion during horizontal channel deformations.

Response 3: We add your recommendations in our discussion in line 496-504 as follows:

In order to consider the analytical results on the correlation between soil erosion rate and all input factors in the RUSLE model by using SPSS, the LS factor is the strongest influence factor of soil erosion in the study area. Nonetheless, the analytical results may not be quite effective because the LS factor varies greatly in the river basin against other factors. Therefore, this section should be considered by regarding the different geological and geomorphological characteristics of the river basin such as mountains, piedmont, and lowland. Moreover, different altitudinal conditions also are important conditions that affect directly the RUSLE input factors. This issue needs to be improved correctly for understanding the influence factor of soil erosion in each feature of the river basin.

Point 4: Please almost everywhere in the text of your manuscript use the “Lancang – Mekong River basin” instead of “Lancang – Mekong River”, otherwise the phrases “Most of the soil erosion in Lancang – Mekong River ...”, “the local soil properties in Lancang – Mekong River ”,“ In Lancang – Mekong River, the highest elevation areas are identified by the highest K values ... ”,“ ... the agricultural activities in Lancang – Mekong River ...”, etc. are extremely incorrect.

Response 4: We revised completely according to your concern in word “Lancang-Mekong River basin”.

Point 5: In the Conclusion, it is advisable to give SPECIFIC practical recommendations in a more concise form, which follow from the results of your study.

Response 5: We revised the conclusion by cutting off and adding some sentences to make it fit.

The RUSLE model is integrated with GIS techniques in this research to assess soil erosion and sediment yield in Lancang–Mekong River basin. The impact of soil erosion on hydropower dams is also considered. The findings indicate that soil erosion occurs in all areas of Lancang–Mekong River basin, accounting for 5,350 t/km2/y of its average soil erosion rate or about 45% of the basin. The north part of upper Mekong River basin and some parts of Thailand have higher terrains than the other areas, and they have good vegetation cover and support practice. Furthermore, the LS factor shows that this factor is the strongest influence factor of soil erosion in the study area. The spatial distribution of soil erosion also indicates that the north part of upper Mekong River basin and the central and south parts of lower Mekong River basin are the most vulnerable areas in terms of increased soil erosion rates due to the movement of sediments to the river. Hence, the dams in this river are highly threatened by sediment problems.

The values of pursuing research on the sediment capacity of each sub-basin of Lancang–Mekong River basin can be summarized as follows. The size of sub-basins and their elevation directly affect the sediment capacity of the river. Moreover, the spatial distribution of sediment deposition and erosion indicates that relatively high sediment erosion occurs along the flow direction of the mainstream, from the north part of upper Mekong River basin to the Mekong Delta. The findings on sediment yield estimation from the modified RUSLE model and the observed sediment data are in good agreement and have high correlation. The proposed technique can be applied in the assessment of sediment yield capacity and sediment deposition in Lancang–Mekong River basin.

The modified RUSLE method is successfully applied to the assessment of the amount and spatial distribution of soil erosion and sediment deposition in Lancang–Mekong River basin. The method can be applied not only to this river but also to other important areas. This study can help policymakers and relevant organizations improve their decision making based on the provided valuable information on soil erosion and sedimentation in this region.

Point 6: Table 4. For almost half of the analysed sub-basins, you obtained a quite good comparability of the observed and calculated SSY-values (10% or less). For others, these values are greater, up to 29%. If it is possible, try to explain the reasons for the relatively large discrepancies in the second case. Maybe this is somehow associated with the environmental features of these river basins (Nam Songkhram, Nam Mun, Se Bang Fai, Nam Ngum, etc.)?

Response 6: We explain the comparability of the observed and calculated SSY-values in line 549-569 as follows:

The result in Table 4 shows quite a good comparability of the observed and estimated SSY from the RUSLE model. Almost half of the sub-basins are approximately 5-10% of percentage error while the remaining sub-basins are estimated at more than 10% from observed values. Sub-basins have a high sediment quantity. The modified RUSLE model can be a well-known simulation. Conversely, if sub-basins have a low sediment quantity, the model shows low performance for sediment yield estimation. These causes may occur from two important factors, including the spatial resolution of the RUSLE input factors and the features of the river basins. For the spatial resolution in analysis, this study chooses rather the coarse grid (5 km) resolution despite the limitations of input data sources. The model can be well-captured in some specific areas from the influence of grid resolution. If this study can be applied to the spatial resolution of 1 km, the sediment yield estimation may be improved efficiently [20]. Meanwhile, the features of the river basins affect directly the sediment yield estimation, especially rainfall from changing climate and land-use change from human activity. Most lands of sub-basins, which are the greater values of percentage error (10-29%), have been changed from the forest areas to the agricultural areas and others, especially Nam Chi, Nam Mun, Nam Songkhram, and Nam Ngum. This issue causes the analysis of the C and P factors to be inaccurate because the C factor was considered from the MODIS satellite image by using the remote sensing techniques, and the P factor was also estimated from the C factor [85]. Furthermore, sub-basins, which are overestimated values, have the features without high slopes when comparing with other sub-basins. Hence, the modified RUSLE model may be able to consider areas with better slopes, which is quite consistent with [20]. Totally, these factors may be the causes of the problem in the study of sediment yield estimation in Lancang-Mekong River basin.

Point 7: Table 2. For the level “Water”, soil loss is more than 8000? I think you had this in mind for the level of “Extreme erosion”.

Response 7: Yes. We have revised it already.

Point 8: It would be interesting for the reader to learn (a few sentences in the Introduction) how (quantitative information) the rates of erosion and sediment yield in the Mekong River basin have changed over the past decades due to changes in climate and land use. This also directly determines the relevance of your research.

Response 8: We add the information in line 62-69 as follows:

In the last few years, Lancang-Mekong River basin has been eroded at an average rate of 5,000 t/km2/y [33], which is moderate erosion level, and it tends to increase in intensity continuously from climate change and land-use change. Conversely, sediment yield in the river basin is decreasing from 250 t/km2/y to 209 t/km2/y because the sediment quantity is trapped by hydropower dams. Historical sediment load (1960-2013) from China to the lower Mekong River indicates clearly that the amount of sediment loads heavily decreases from 84.7 Mt/y to 10.8 Mt/y and 147 Mt/y to 66 Mt/y at Chiang Saen and Pakse stations, respectively [37-39].

Point 9: There are several typos in the text. For example, line 547 “... that Mekong Delta is the most vulnerable rea in terms of risk of soil loss …”. Please check the typos everywhere in the text.

Response 9: We have already completed for checking according to your recommendations.

Round 2

Reviewer 1 Report

The Authors made good improvement in the paper. I think that still the part on Dam sediment trapping must be further explained.

Also the paper is unnecessary too long and must be revised in this sense.

Overall I encourage the Authors to make the effort in order to have a good paper.

Author Response

Thank you very much for your kind review of our manuscript. We would like to explain according to your kind recommendations.

Point 1: The Authors made good improvement in the paper. I think that still the part on Dam sediment trapping must be further explained.

Also the paper is unnecessary too long and must be revised in this sense.

Overall I encourage the Authors to make the effort in order to have a good paper.

Response 1: We would like to explain as follows:

Firstly, we rewrite and add some sentences in second paragraph in the section of soil erosion impact on dams (line 594-618), which is shown below. Besides, we attempt to describe clearly regarding the soil erosion modulus, which affects the storage capacity of dams directly.

The impact of soil erosion on dams in Lancang–Mekong River’s mainstream can be analysed in two parts based on the water sources of the river, namely, the upper Mekong River (with three river areas from Lancang basin) and the lower Mekong River (composed of the northern highlands, Khorat Plateau, Tonle Sap, and Mekong Delta). The upper Mekong River basin covers 180,000 km2 or approximately 24% of the study area, while the lower Mekong River basin covers 570,000 km2 or approximately 76%. The soil erosion modulus of the upper Mekong River basin is 235.7 t/km2/y. Its percentage relative to total soil erosion is approximately 36% even if this area is smaller than the lower Mekong River basin. The soil erosion modulus of the lower Mekong River basin is 198.2 t/km2/y, which represents approximately 64% of the total occurrence of soil erosion. The results of the soil erosion modulus can be explained that the reservoirs located in the upper Mekong River basin are more vulnerable from soil erosion and increased sediment. Consequently, dams are likely to be at risk of decreasing storage capacity continually. Our results are consistent with the findings of past studies on the impact of soil erosion on dams and sediment trapping. For instance, [47] reported that the sediment trapping rates of dams under construction and the planned dams in Lancang–Mekong River basin will increase from 51% to 69% due to the high heterogeneity of specific sediment yield in the different parts of the basin, and much higher trapped sediment load is predicted because of soil erosion resulting from socio-economic development. More than 50% of the sediment load (approximately 140 Mt) in Lancang–Mekong River basin is expected to be trapped annually. Besides, more than 60% of the sediment load originates from China’s end of Lancang–Mekong River’s mainstream. Existing dams, dams under construction, and planned dams are expected to have the highest impact on storage capacity due to sediment load. [28] reported that main dams in Lancang River, such as Manwan, Gongguoqiao, Dachaoshan, and Jinghong, have sediment trapping rates between 30% and 70% because of the high sediment yield in Lancang–Mekong River’s mainstream and sub-basins. The storage capacity of reservoirs will be continuously decreased by sediment load due to soil erosion in the reservoir upstream.

Finally, we attempt to cut off some sentences from our paper and revise some paragraphs according your kind recommendations to improve to be good paper.

Reviewer 3 Report

In general, the authors quite satisfactorily answered my comments. As a reviewer, I recommend the manuscript for publication in the journal "Water".

Author Response

Point 1: In general, the authors quite satisfactorily answered my comments. As a reviewer, I recommend the manuscript for publication in the journal "Water".

Response 1: Thank you very much for your kind recommendation of our manuscript to publish in the journal “Water”.
